# Quality of Life, Perceived Social Support, and Treatment Adherence Among Methadone Maintenance Program Users: An Observational Cross-Sectional Study

**DOI:** 10.3390/healthcare13151849

**Published:** 2025-07-29

**Authors:** Pedro López-Paterna, Ismail Erahmouni-Bensliman, Raquel Sánchez-Ruano, Ricardo Rodríguez-Barrientos, Milagros Rico-Blázquez

**Affiliations:** 1La Velada Healthcare Centre, Área de Gestión Sanitaria Campo de Gibraltar Este, Servicio Andaluz de Salud, 11300 La Línea de la Concepció, Spain; 2Algeciras Norte Healthcare Centre, Área de Gestión Sanitaria Campo de Gibraltar Este, Servicio Andaluz de Salud, 11205 Algeciras, Spain; 3Research Unit, Primary Care Assistance Management, Madrid Health Service, 28035 Madrid, Spain; 4Research Network on Chronicity, Primary Care and Health Promotion-RICAPPS-(RICORS), Instituto de Salud Carlos III (ISCIII), 28003 Madrid, Spain; 5Gregorio Marañon Health Research Institute, Madrid Health Service, 28007 Madrid, Spain; 6Nursing Department, Faculty of Nursing, Physiotherapy and Podiatry, Complutense University of Madrid, 28040 Madrid, Spain

**Keywords:** quality of life, methadone, substance-related disorders, addictive behaviour, social support, treatment adherence and compliance, community health services, public health, social determinants of health, vulnerable populations

## Abstract

**Background/Objectives:** The consumption of opioids is a public health problem that significantly affects quality of life. In Spain, 7585 people are enrolled in the Methadone Maintenance Programme (MMP), which is an effective intervention with a low adherence rate. In this study, factors associated with the quality of life of MMP users, especially perceived social support and treatment adherence, were analysed. We hypothesised that low levels of adherence and social support would be associated with poorer quality of life. **Methods:** This was a cross-sectional observational study with an analytical approach. Quality of life (WHOQoL-BREF), perceived social support (DUKE-UNC-11), and treatment adherence (MMAS-8) among MMP users were studied, and data on sociodemographic and clinical characteristics were collected through ad hoc questionnaires and a review of electronic medical records. Linear and logistic regression models were used. **Results:** A total of 70 individuals were included in this study. The mean age was 56.9 years, and 83% of the participants were male. The perceived quality of life was low in the four domains evaluated (range of 47.4–48.2). A total of 38.57% of the participants had low perceived social support. Treatment adherence was low or moderate in 77.1% of the participants. Greater perceived social support was associated with better quality of life in all domains (*p* < 0.05). Quality of social life was negatively associated with the use of nonbenzodiazepine neuroleptics and HIV status. Treatment adherence was lower in insulin therapy users. **Conclusions:** Social support is a key determinant of the quality of life of MMP users. Health policies should promote social support networks as a strategy to improve the well-being of this population.

## 1. Introduction

The use of opioids constitutes a large public health problem, with clinical, social, and economic consequences. There were an estimated 60 million heroin and opioid users worldwide in 2022, 30 million of whom used heroin. In 2020, the prevalence of high-risk users was estimated to be 860,000 in Europe alone [1,2]. Opioid use disorder is associated with high premature morbidity and mortality [3]. The World Health Organization estimates approximately 125,000 deaths from opioid use per year [4].

In Spain, the prevalence of heroin use has remained at 0.1% since 1995 [5]. Heroin use tends to coexist with mental disorders, infectious diseases, and polydrug use, which increases the clinical, psychological, and social vulnerability of these individuals [6,7,8]. In addition, factors such as social exclusion, unemployment, and crime reflect the multidimensional impact of the problem. The approach to this problem entails a high cost, not only for the public health system but also for social, penitentiary, and administrative services [9,10]. Reintegrating this population into society and the workforce is usually a process that requires a long period of readjustment due to polyconsumption problems, less social support, less economic means, and criminal records [11,12].

The Methadone Maintenance Programme (MMP) is the main drug replacement strategy recommended for the treatment of opioid use disorders. The effectiveness of this programme has been widely demonstrated, not only in reducing consumption but also in reducing morbidity and mortality, the transmission of infectious diseases, and crime rates. However, the treatment involves more than just prescribing methadone and requires a comprehensive approach that considers the associated risks, such as clinical complications, relapses, the psychosocial sphere, and quality of life [13,14,15].

In this sense, psychosocial support is an essential component of the MMP. Therapeutic adherence is a challenge, especially in the initial phases, owing to factors such as stigma, the obligation to obtain medication daily, and interference with daily life [16]. Anxiety is one of the most frequently occurring symptoms during treatment, involving frequent psychiatric comorbidities and an elevated risk of suicide. Factors such as a low educational level, the use of benzodiazepines, or unemployment increase the risk of anxiety [17,18].

Quality of life has become a key indicator of health, and its measurement in MMP users makes it possible to assess the impact of interventions beyond the control of consumption [19]. A higher degree of life satisfaction is associated with greater adherence to treatment. However, individuals with opioid use usually have a low level of social support caused by the loss of family ties, friendships, or financial aid [20,21,22]. This lack of support increases the risk of treatment abandonment and increases the vulnerability of people in the MMP at the personal, social, and economic levels [21]. Opioid users are among the most marginalised groups that use illegal substances [2].

No evidence linking the three outcome variables of this study within the MMP population has been found; however, associations among them have been documented in other population groups [23,24]. Theoretical frameworks such as the stress-buffering hypothesis suggest that social support can mitigate the negative effects of chronic stress on health outcomes. Additionally, behavioural models emphasise the influence of social and contextual factors on health behaviours like treatment adherence. Empirically, associations have been found between social support and treatment adherence [25,26], as well as between social support and quality of life [27,28]. This provides a conceptual and empirical foundation to investigate such an association in MMP users.

Despite the impact of opioid consumption and its high health care and social burdens, in Spain, there is little scientific evidence that addresses the quality of life in this population and the relationship of quality of life with social support and therapeutic adherence. Although international studies have been conducted, their applicability to the Spanish context is limited due to sociocultural differences.

In 2022, 7585 people were admitted to the MMP in Spain. Of these individuals, 2089 (27.3%) were from the province of Cádiz, which represents only 2.65% of the total Spanish population [29,30,31,32]. This marked disparity highlights the need for targeted interventions and region-specific strategies in Cadiz.

Therefore, the objective of this study was to explore the relationships among quality of life, perceived social support, and treatment adherence in a sample of MMP users in Spain. We hypothesised that this population would have low adherence and perceived social support, which would be significantly associated with lower quality of life across all domains. The results are intended to provide evidence to guide future community interventions focused on the overall well-being of this population.

## 2. Materials and Methods

### 2.1. Design

This cross-sectional observational study was carried out with an analytical approach in the Campo de Gibraltar region (Spain) at the health centres responsible for dispensing the substitution treatment (methadone).

The inclusion criteria were individuals older than 18 years, individuals who had participated in the MMP for at least 6 months at the beginning of this study, individuals who attended the medication collection meeting in person and on time, and individuals who signed the informed consent form. People with an acute episode of mental illness, cognitive impairment, or language barriers that prevented them from providing informed consent or completing the questionnaires were excluded.

No formal sample size calculation or probability sampling was performed since participation was offered to all users who met the eligibility criteria. Ultimately, 70 people were included in this study. Figure 1 represents the flow chart of this selection process.

The STROBE checklist is available as Appendix A.

### 2.2. Variables and Information Collection

Data were collected through self-administered questionnaires in health centres between July 2024 and March 2025. Three scales that have been validated in Spain [33,34,35] were used to measure the main outcome variables: (I) The World Health Organization Quality of Life Instrument—Short Version (WHOQoL-BREF) is a 26-item instrument that measures quality of life across four domains: physical health, psychological well-being, social relationships, and the environment. Each domain is scored on a scale from 0 to 100, with higher scores indicating better perceived quality of life [36]. (II) The Morisky Medication Adherence Scale (MMAS-8) is an 8-item scale that measures self-reported medication adherence. Total scores range from 0 to 8, with higher scores indicating better adherence, which is categorised as follows: high adherence (score = 8), medium adherence (score of 6–8), and low adherence (score < 6) [37]. (III) The Duke–UNC Functional Social Support Questionnaire (DUKE-UNC-11) is an 11-item scale that evaluates the perception of functional social support. Total scores range from 11 to 55, and lower scores indicate lower perceived social support. In Spain, a score below 32 is considered indicative of low social support [38].

In addition, an ad hoc questionnaire (Appendix B) was used to collect data on sociodemographic variables (age, sex, educational level, marital status, number of children, employment status, nationality, coexistence status, and prison history) and a variable related to the treatment (duration of participation in the MMP). The coexistence variable was recategorised as “alone”, “family”, and “sharing a home”.

Information on drug treatment (dispensing frequency, methadone dose, and use of benzodiazepines, antihypertensives, neuroleptic anxiolytics, inhalers, gastric drugs, oral antidiabetics, insulin therapy, nonopioid analgesia, opioid analgesia, and/or antiplatelet drugs) and comorbidities (hepatitis C, hepatitis B, human immunodeficiency virus (HIV), and cardiac, mental, respiratory, oncological, and/or neurological pathologies) was extracted from electronic medical records.

### 2.3. Statistical Analysis

A descriptive analysis of the sociodemographic and clinical characteristics of the population was carried out using absolute and relative frequencies for qualitative variables and measures of central tendency and dispersion (mean and standard deviation) for quantitative variables. Chi-square tests were used to compare proportions to study the associations between variables, and Student’s *t*-tests or ANOVA were used to compare means according to the number of groups.

Linear regression models adjusted for age and sex were developed to study the possible associations of sociodemographic and clinical variables with quality of life and therapeutic adherence. A logistic regression model adjusted for age and sex was developed to analyse the possible associations of sociodemographic and clinical variables with perceived social support (dichotomised into “low” and “normal” according to the validated cut-off point). The magnitude of the association is presented as the beta coefficient and odds ratio (OR), with their respective 95% confidence intervals. The selection of variables for each model was based on those that showed statistically significant associations in the bivariate analyses and those considered clinically relevant on the basis of previous evidence or expert judgement.

Multicollinearity between the independent variables included in the models was not examined, as each variable provides distinct information, allowing us to isolate the individual contribution of each factor.

No imputation was necessary, as the primary variables of interest were complete. Complete-case analysis was used.

Stata 15 statistical software was used in this study.

## 3. Results

A total of 70 people, with a mean age of 56.9 years (7.61), participated in this study. A total of 83% of the participants were male. The sociodemographic characteristics of the sample are presented in Table 1. The unemployment rate was high (90%), and 60% of the participants had prison records. The most common educational level was primary education (32.86%), and 41.43% of the participants were single.

In terms of the duration of treatment, 36.23% of the participants had a duration of participation in the MMP between 20 and 30 years. The most prevalent comorbidities were hepatitis C (44.93%), cardiovascular disease (24.64%), and HIV (17.39%). The most common pharmacological treatment used was benzodiazepines (68.12%), followed by neuroleptic anxiolytics (34.78%). Table 2 summarises the clinical characteristics of the participants.

The general items of the WhoQoL-BREF indicated a global perception of quality of life and moderately low satisfaction with the state of health (Figure 2 and Figure 3). In the specific domains, the mean values ranged from 47 to 48 points (Figure 4). Nevertheless, the general perception of quality of life was rated at least as “neither poor nor good” by 67.1% of the participants, which contrasts with the results obtained in the specific domains of the scale. In terms of the state of health, 41.4% of the participants reported that they were “dissatisfied” or “very dissatisfied”, whereas only 24.2% reported that they were “satisfied” or “very satisfied”. Most of the participants perceived a moderate level of social support, although 38.57% perceived a low level of social support (Figure 5). Regarding therapeutic adherence (MMAS-8 score), 15.71% of the participants had low treatment adherence, and 61.43% had moderate treatment adherence (Figure 6).

Linear and logistic regression models adjusted for age and sex were developed to analyse the factors associated with the main variables (quality of life, perceived social support, and treatment adherence). The results are presented in Table 3, Table 4, Table 5, Table 6, Table 7 and Table 8.

Higher perceived social support was significantly associated with higher scores in the physical domain of quality of life (8.12; 95% CI 0.51; 15.72). In addition, the use of inhalers and antiplatelet agents was negatively associated with the scores in this domain (Table 3).

Perceived social support was significantly associated with a higher score in the psychological domain of quality of life (13.08; 95% CI 4.47; 21.69). The presence of a mental pathology showed a nonsignificant positive association with perceived social support (Table 4).

In the social domain, perceived support was positively associated with quality of life (17.78; 95% CI 9.32; 26.23). In contrast, the use of nonbenzodiazepine anxiolytics and neuroleptics was associated with a lower quality of social life, as was HIV status (Table 5).

The level of perceived social support was significantly associated with a higher score in the environmental domain of quality of life (12.97; 95% CI 6.22; 19.72) (Table 6).

A negative association was observed between insulin therapy use and therapeutic adherence (−0.65; 95% CI −1.20; −0.10), indicating lower adherence in people with diabetes treated with insulin (Table 7).

The type of coexistence was significantly associated with the level of perceived social support. Sharing a home without family ties was associated with a greater probability of having perceived social support (OR = 3.84; CI95% 1.07;13.83). Living with relatives and employment showed no association (Table 8).

## 4. Discussion

A lower quality of life among the participants was observed in the present study, with lower values in all its domains in the study population than in the general Spanish population. The latter presents physical, psychological, social, and environmental quality of life scores of 90.38, 81.94, 81.06, and 63.75, respectively [39]. Prospective cohort studies conducted in China and Norway have also shown that individuals enrolled in the MMP report quality of life scores below the national average [40,41]. These results are consistent with those of other studies conducted in different sociocultural contexts [15,19,22,42]. The perceived social support score was 26% lower than that in the general population, with an average of 33.86 points versus 48.21 points [43]. According to the results of other studies [44], this difference suggests a possible lack of support networks and social resources among MMP users, which could affect their therapeutic process and their overall well-being. Regarding therapeutic adherence, only 22.9% of the participants presented adequate adherence, whereas 49.9% of the general Spanish population had adequate adherence [45]. This figure is notably lower than that reported in other studies, such as transversal studies carried out in Vietnam, where an optimal adherence rate of 34.4% was observed [46].

In terms of sociodemographic elements, the unemployment rate was 90% for the MMP users in this study, which was much higher than the rate of 59.3% reported for MMP users in Andalusia [47]. These unemployment rates are also substantially higher than those reported in other studies, such as a prospective cohort study conducted in China, which reported an unemployment rate of 70%, a cross-sectional study carried out in Myanmar, which reported a rate of 8.7%, and a study conducted in Vietnam, which reported a rate of 7% [22,40,46]. This difference could be explained in part by the local context of the study, which was carried out in the towns of La Línea de la Concepción and Algeciras, with unemployment rates of 32.39% and 25.68%, respectively, compared with the national average of 10.61% and the Andalusian average of 15.76. Both municipalities are among the Spanish localities with the highest unemployment rates among those with more than 45,000 inhabitants [48,49]. Cross-sectional studies conducted in other sociocultural contexts have established an association between unemployment and lower quality of life [50,51]. The profile of an MMP user is a male of approximately 44 years of age who has previously received some type of substitution treatment, with a primary education level, in a situation of unemployment, and who goes to treatment of his own free will [29,47]. In terms of coexistence, no differences were observed with respect to the MMP population of Andalusia (21.4% of participants lived alone and 61.8% coexisted) [47]. Sixty per cent of the sample had a prison record, a figure that contrasts sharply with the 2.98% estimated for individuals in the general Spanish population who served prison time between 2009 and 2019 [52,53]. Substance use has been identified as a risk factor for criminal conduct [12,52].

Among the comorbidities, hepatitis C and HIV stood out. Among the participants, 44.29% tested positive for hepatitis C, whereas 1.7% of the general population in Spain tested positive for hepatitis C [54]. The prevalence of HIV was 17.14%, whereas it was 0.32% at the national level [55]. In another cross-sectional study conducted in Myanmar, even higher rates were reported: 47.97% for hepatitis C and 37% for HIV [22]. Injecting drug use, poor health habits, and poor health conditions could explain this exponential increase [54].

All these factors are related to quality of life, so they must be accounted for when interpreting the results [53,56,57].

In a descriptive analytical study carried out in a Spanish population (*n* = 523) with a mean age of 46.67 years (compared to 56.94 years in our study) and 80.5% of male participants (similar to 83% in this study), the scores were greater in terms of quality of life [58]. The scores in the aforementioned studies were 57.38 for physical quality of life, 51.63 for psychological quality of life, 47.38 for social quality of life, and 54.19 for environmental quality of life [59]. These differences may be due to the sample size of both studies, as the study was carried out mainly in Madrid, whose socioeconomic situation is different, or the age difference between the two samples [60]. In other countries with MMP populations, such cross-sectional studies have reported greater quality of life scores across all domains than those of the population in our study [22,61].

Other studies have reported a relationship among the duration of participation in the MMP, dosage, and quality of life; however, this relationship could not be examined in our study because of the sample size [15,62,63].

Perceived social support was revealed as a key factor throughout the study. As reflected by the linear regression models, perceived social support is positively associated with all domains of quality of life. In the regression model for physical quality of life, perceived social support, antiplatelet treatment use, and inhaler use were associated with each other. It was decided to use different types of pharmacological treatment as independent variables instead of the base pathologies, since they did not allow us to discriminate the clinical severity, which limited their explanatory use. These medications are commonly used by individuals with respiratory or cardiac conditions, which are known to be associated with poorer quality of life [64,65]. In the psychological domain, perceived social support and the presence of mental illness are associated with quality of life. Although the positive association between mental disorders and quality of life may seem surprising, it could be explained by the sample size, active treatment for these conditions, and the adjustment of health-related quality of life expectations [66]. The quality of social life, perceived social support, the use of nonbenzodiazepine/neuroleptic anxiolytics, and the diagnosis of HIV were related to this domain. The negative association of drugs could be due to the effect of the underlying pathology on social relationships. On the other hand, the social stigma associated with HIV can contribute to a greater perception of social isolation [67]. In the environmental domain, perceived social support was the only variable with a significant association. The logistic regression model for perceived social support revealed a greater probability of adequate perceived social support among those who shared a home with nonfamily members and among those who were employed. This last finding was included because of its clinical relevance. These results coincide with those of previous studies that indicate the positive effects of employment and coexistence on the subjective perception of social support [68,69].

Although perceived social support and quality of social life seem to overlap, they actually belong to different dimensions. Perceived social support refers to “the availability and subjective adequacy of social connections” [70], whereas the quality of social life refers to “an individual’s perception of their life situation, since, in the context of their culture and value systems and in relation to their objectives, expectations, standards and concerns regarding their social relationships, their social support and their sexual sphere” [71]. These results support the development of integrated community interventions combining pharmacological treatment with peer support, job training, and stigma-reduction strategies, particularly for those with comorbidities such as HIV or mental health disorders.

### Strengths and Limitations of the Study

The main limitation of this study was the lack of attendance at the referral centre on the day of methadone dispensing (29.07% of the potential sample). To overcome this limitation, multiple follow-up contacts were made with each centre via the reference nurses. In addition, data collection coincided with medication delivery, preventing additional travel for the participants. Another limitation was the proportion of users (14.28%) whose methadone was obtained by a family member.

One of the limitations of this study is the obligatory nature of adherence to treatment for the MMP users. This may have conditioned their responses, leading to desirability bias. To reduce this risk, the questionnaires were not carried out while the reference nurses were present at the dispensation; rather, the MMP users were taken to a separate room.

In addition, there are limitations inherent to the use of secondary sources, such as electronic medical records. Data on the main outcome variables were collected to minimise the impact.

Finally, the absence of a formal sample size calculation and the relatively modest sample of 70 participants included may limit the generalisability of our findings. Possible selection and information biases are acknowledged as methodological limitations. We recommend that future studies consider including larger, multicentre samples to allow for broader comparisons and subgroup analyses.

The main strength of this study is that the entire accessible population of the participating areas was invited to participate, accounting for 3.4% of the total number of people enrolled in the MMP in Spain [72] and 12% of the total number of MMP users in Andalusia [47].

## 5. Conclusions

The findings of this research can contribute to decision-making regarding public health policies, allowing the optimisation of intervention strategies aimed at the population enrolled in the MMP and focusing attention on the most vulnerable aspects identified. In particular, perceived social support has been shown to significantly affect quality of life across all domains. As such, intervention strategies should incorporate social and community-based components, including active employment support, vocational training, and initiatives to foster social networks and reduce stigma. Additionally, multidisciplinary approaches that include psychological support and address adherence to treatment may improve outcomes in this population. These findings provide valuable insight for tailoring public health responses to better meet the complex needs of people enrolled in the MMP.

## Figures and Tables

**Figure 1 healthcare-13-01849-f001:**
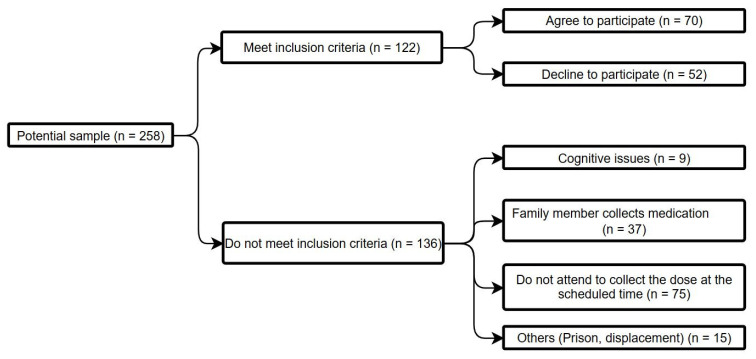
Flow chart.

**Figure 2 healthcare-13-01849-f002:**
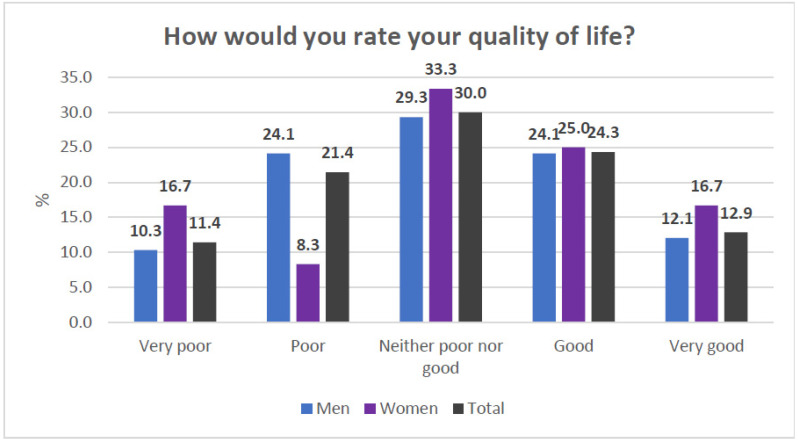
Item 1 of the WhoQoL-BREF scale.

**Figure 3 healthcare-13-01849-f003:**
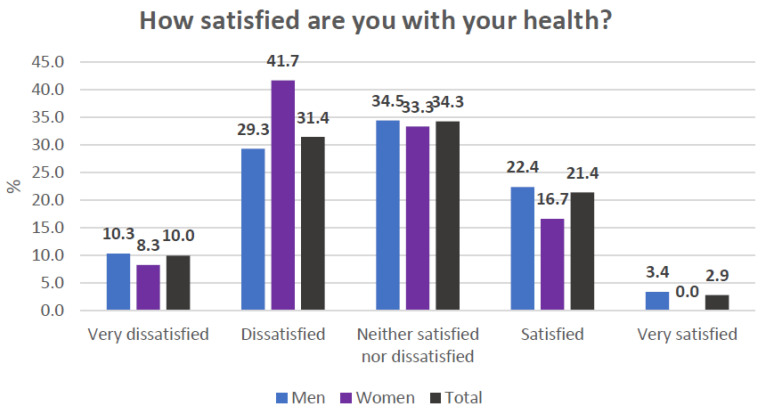
Item 2 of the WhoQoL-BREF scale.

**Figure 4 healthcare-13-01849-f004:**
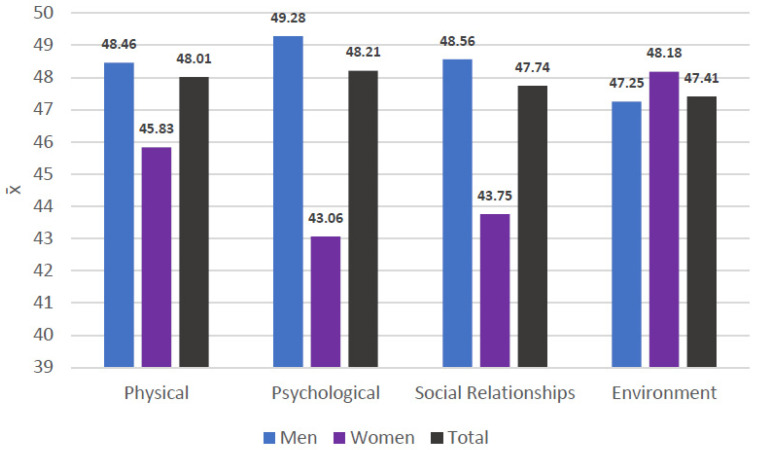
Quality of life of the participants (WhoQoL-BREF).

**Figure 5 healthcare-13-01849-f005:**
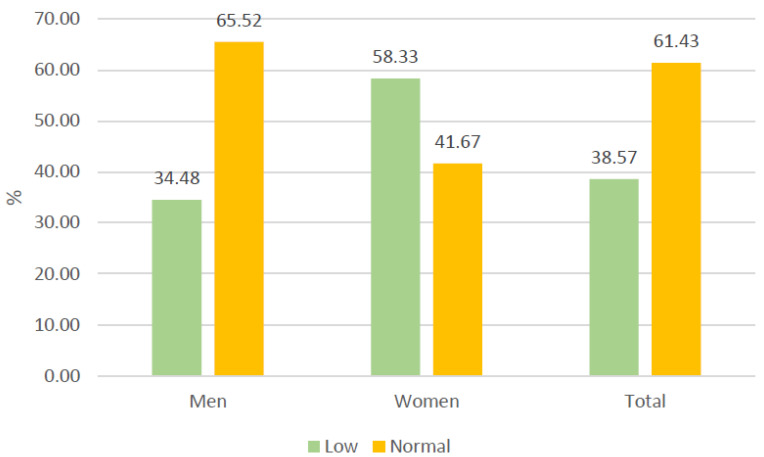
Perceived social support of the participants (Duke-UNC-11).

**Figure 6 healthcare-13-01849-f006:**
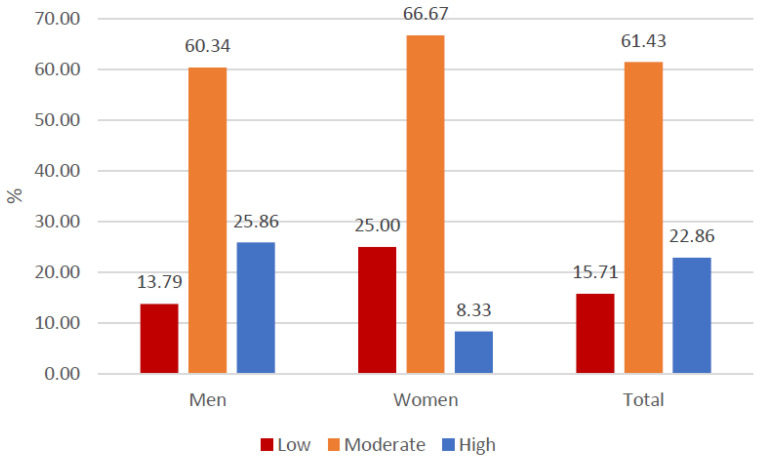
Treatment adherence of the participants (MMAS-8).

**Table 1 healthcare-13-01849-t001:** Sociodemographic characteristics.

	Men (*n* = 58; 83%)	Women (*n* = 12; 17%)	Total (*n* = 70)
Age x̄ (SD)	57.34 (7.79)	55 (6.59)	56.94 (7.61)
Educational level-*n* (%)			
Did not finish school	14 (24.14%)	2 (16.67%)	16 (22.86%)
Primary education	17 (29.31%)	6 (50%)	23 (32.86%)
Secondary education	17 (29.31%)	3 (25%)	20 (28.57%)
Baccalaureate	5 (8.62%)	1 (8.33%)	6 (8.57%)
University	5 (8.62%)	0 (0%)	5 (7.14%)
Marital status-*n* (%)			
Single	27 (46.55%)	2 (16.67%)	29 (41.43%)
Married	4 (6.9%)	1 (8.33%)	5 (7.14%)
Divorced/Separated	12 (20.69%)	2 (16.67%)	14 (20%)
Widowed	4 (6.9%)	1 (8.33%)	5 (7.14%)
Has a partner	11 (18.97%)	6 (50%)	17 (24.29%)
Occupation-*n* (%)			
Employed	5 (8.62%)	2 (16.67%)	7 (10%)
Unemployed	53 (91.38%)	10 (83.33%)	63 (90%)
Number of children-*n* (%)			
0	19 (32.76%)	2 (16.67%)	21 (30%)
1	14 (24.14%)	3 (25%)	17 (24.29%)
2	15 (25.86%)	3 (25%)	18 (25.71%)
3	6 (10.34%)	0 (0%)	6 (8.57%)
4	1 (1.72%)	2 (16.67%)	3 (4.29%)
5 or more	3 (5.17%)	2 (16.67%)	5 (7.14%)
Prison history-*n* (%)			
Yes	38 (65.52%)	4 (33.33%)	42 (60%)
No	20 (34.48%)	8 (66.67%)	28 (40%)
Nationality-*n* (%)			
Spanish	57 (98.28%)	12 (100%)	69 (98.57%)
German	1 (1.72%)	0 (0%)	1 (1.43%)
Coexistence-*n* (%)			
Alone	19 (32.76%)	2 (16.67%)	21 (30%)
With a partner	8 (13.79%)	6 (50%)	14 (20%)
With a partner and children	1 (1.72%)	0 (0%)	1 (1.43%)
With children	1 (1.72%)	0 (0%)	1 (1.43%)
Shares a house	29 (50%)	3 (25%)	32 (45.71%)
Street situation	0 (0%)	1 (8.33%)	1 (1.43%)

**Table 2 healthcare-13-01849-t002:** Clinical characteristics.

	Men (*n* = 57; 82.61%)	Women (*n* = 12; 17.39%)	Total * (*n* = 69)
Duration of participation in the MMP-*n* (%)			
<1 year	3 (5.17%)	2 (16.67%)	5 (7.25%)
1–5 years	5 (8.62%)	1 (8.33%)	6 (8.70%)
5–10 years	6 (10.34%)	1 (8.33%)	7 (10.14%)
10–20 years	11 (18.97%)	2 (16.67%)	13 (18.84%)
20–30 years	22 (37.93%)	3 (25%)	25 (36.23%)
>30 years	11 (18.97%)	3 (25%)	14 (20.29%)
Methadone treatment dose in mg/day—median (IQR)	50 (30–80)	40 (17.5–70)	45 (30–80)
Delivery interval between doses in days—median (IQR)	7 (7–14)	7 (1–7)	7 (7–14)
1 day, *n* (%)	7 (12.28%)	4 (33.33%)	11 (15.94%)
7 days, *n* (%)	28 (49.12%)	6 (50%)	34 (49.28%)
14 days, *n* (%)	19 (33.33%)	2 (16.67%)	21 (30.43%)
28 days, *n* (%)	3 (5.26%)	0 (0%)	3 (4.35%)
Comorbidities-*n* (%)			
Hepatitis C	27 (47.37%)	4 (33.33%)	31 (44.93%)
Hepatitis B	1 (1.75%)	0 (0%)	1 (1.45%)
HIV	11 (19.30%)	1 (8.33%)	12 (17.39%)
Cardiovascular disease	16 (28.07%)	1 (8.33%)	17 (24.64%)
Mental pathology	11 (19.30%)	3 (25%)	14 (20.29%)
Respiratory pathology	9 (15.79%)	2 (16.67%)	11 (15.94%)
Musculoskeletal pathology	10 (17.54%)	1 (8.33%)	11 (15.94%)
Oncological pathology	3 (5.26%)	0 (0%)	3 (4.38%)
Neurological pathology	4 (7.02%)	0 (0%)	4 (5.80%)
Pharmacological treatments-*n* (%)			
Benzodiazepines	41 (71.93%)	6 (50%)	47 (68.12%)
Antihypertensives	12 (21.05%)	1 (8.33%)	13 (18.84%)
Neuroleptic anxiolytics	20 (35.09%)	4 (33.33%)	24 (34.78%)
Inhalers	14 (24.56%)	2 (16.67%)	16 (23.19%)
Gastric drugs	15 (26.32%)	4 (33.33%)	19 (27.54%)
Oral antidiabetics	6 (10.53%)	3 (25%)	9 (13.04%)
Insulin therapy	3 (5.26%)	2 (16.67%)	5 (7.25%)
Non-opioid analgesia	13 (22.81%)	4 (33.33%)	17 (24.64%)
Opioid analgesia	7 (12.28%)	0 (0%)	7 (10.14%)
Platelet antiaggregant	7 (12.28%)	0 (0%)	7 (10.14%)

* One participant did not have an electronic medical record.

**Table 3 healthcare-13-01849-t003:** Physical quality of life (linear regression model).

	CRL *	95% CI **	*p*
Perceived social support	8.12	0.51; 15.72	**0.037**
Platelet antiaggregant	−11.32	−23.36; 0.71	0.065
Inhalers (respiratory therapy)	−9.37	−17.97; −0.77	**0.033**
R^2^ Adjusted = 0.1435			

* Coefficient of linear regression. ** 95% CI: 95% confidence interval. Numbers in Bold means significant values.

**Table 4 healthcare-13-01849-t004:** Psychological quality of life (linear regression model).

	CRL *	95% CI **	*p*
Perceived social support	13.08	4.47; 21.69	**0.003**
Mental pathology	7.68	−2.53; 17.89	0.138
R^2^ Adjusted = 0.1123			

* Coefficient of linear regression. ** 95% CI: 95% confidence interval. Numbers in Bold means significant values.

**Table 5 healthcare-13-01849-t005:** Social quality of life (linear regression model).

	CRL *	95% CI **	*p*
Perceived social support	17.78	9.32; 26.23	**0.000**
Anxiolytics (nonbenzodiazepines)/neuroleptics	−13.64	−22.22; −5.05	**0.002**
HIV	−13.61	−24.26; −2.96	**0.013**
R^2^ Adjusted = 0.2731			

* Coefficient of linear regression. ** 95% CI: 95% confidence interval. Numbers in Bold means significant values.

**Table 6 healthcare-13-01849-t006:** Environmental quality of life (linear regression model).

	CRL *	95% CI **	*p*
Perceived social support	12.97	6.22; 19.72	**0.000**
R^2^ Adjusted = 0.1479			

* Coefficient of linear regression. ** 95% CI: 95% confidence interval. Numbers in Bold means significant values.

**Table 7 healthcare-13-01849-t007:** Treatment adherence (linear regression model).

	CRL *	95% CI **	*p*
Insulin therapy	−0.65	−1.20; −0.10	**0.020**
R^2^ Adjusted = 0.086			

* Coefficient of linear regression. ** 95% CI: 95% confidence interval.

**Table 8 healthcare-13-01849-t008:** Perceived social support *(logistic regression model)*.

	OR *	95% CI **	*p*
Coexistence (family)	0.73	0.18; 2.99	0.661
Coexistence (sharing a home)	3.84	1.07; 13.83	**0.040**
Employment	6.99	0.65; 75.04	0.109
Pseudo R^2^ = 13.56			

* Odds ratio. ** 95% CI: 95% confidence interval.

## Data Availability

The Ethics Committee approved this research without considering the option of data sharing. The data can be requested by contacting the main author. However, each new project on the basis of these data must be previously submitted to the Ethics Committee for approval.

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
