# Peer review of "Quality of Life, Perceived Social Support, and Treatment Adherence Among Methadone Maintenance Program Users: An Observational Cross-Sectional Study"

_healthcare, 2025, doi:10.3390/healthcare13151849_

Round 1

Reviewer 1 Report

Comments and Suggestions for Authors

Thank you for the opportunity to review the manuscript “Quality of life, perceived social support and treatment adherence of the users of the Methadone Maintenance Program: An observational cross-sectional study”

Below are points which need to be addressed in order to improve the presented work to appeal to a wider audience

METARIALS and METHODS

-add full titles/names of questionnaires used as well as appropriate references

-did this research team or any other adapt and validate these questionnaires into Spanish?

-further explain whether electronic health records are readily available to all healthcare providers or the research team needed special approval to access it

-what is the cut-off point for dichotomizing the social support levels? I would suggest to write how all the scales/questionnaires used are scored and what each cut-off point represents or what the scale od each result is

-further expand on what each used questionnaire explores

RESULTS

-a substantial portion of participants are diagnosed with hepatitis and/or HIV can you explain why there is no record of antiretroviral therapy within pharmacological treatment?

-please explain why you chose specific variables for each linear regression model (each health domain). Did you test all variables and chose to represent only several of them or? For example, why was use of platelet antiaggregant used in the physical quality of life regression model but you did not use insulin therapy as a variable? Why wasn’t use of benzodiazepines used as a variable in psychological quality of life regression model?

DISCUSSION

-For statements:

 the environmental domain, perceived social support was the only variable with a

significant association. The scale used includes social elements such as "environment",

"home", "community" or "recreational activities  and The logistic regression model for perceived social support revealed a greater probability of adequate perceived social support among those who share living with nonfamily members and among those who work. This last finding was included for its clinical relevance

I find this very confusing and unfinished. How do these findings reflect on practice? What is the clinical relevance? How does this affect the provision of healthcare for this patients?

-What are the main suggestions you can give as a solution for your findings. This is in regard to your statement in the conclusion that the findings can contribute to decision-making. What would be appropriate intervention strategies?

Reviewer 2 Report

Comments and Suggestions for Authors

The paper entitled "Quality of life, perceived social support and treatment adherence of the users of the Methadone Maintenance Program: An observational cross-sectional study." is based on an important topic in Addiction Biology which is Methadone Maintenance therapy. However, such these studies focusing on the sociopsychological aspects of addiction are crucial for healthcare systems, the sample sizes of such these studies compared to the Genetic studies for example, can be larger than the present work. However, I find some merits in the present paper and have some minor comments need to be carefully addressed by the authors, These issues are listed below:
1- The whole manuscript should be revised by a Native English to improve the readership. 
2-The authors should revise the whole manuscript and decrease the number of paragraphs by merging the related sentences together. 
3- Discussion section needs a great deal of revisions both in its structure (decrease the abundant paragraphs) and adding more related works for comparison. More clearly, the authors should decrease describing their results in the Discussion section and move the additional descriptions of their results in Result section and add more investigations similar to their works. 
4-  The Limitation sub-section requires a deeper insight, the authors should add the limitation of their sample size and suggest more samples in the similar future studies and also adding more clinical data such as dosage of Methadone and Insomnia status. 

Comments on the Quality of English Language

The whole paper should be revised by a Native English for both its structural and its content. 

Reviewer 3 Report

Comments and Suggestions for Authors

Title of the manuscript:

Quality of life, perceived social support and treatment adherence of the users of the Methadone Maintenance Program: An observational cross-sectional study. 

The manuscript submitted addresses a topic of high relevance and actuality, linked to quality of life, perceived social support and adherence to treatment in users of the methadone maintenance program (MMP) in Spain. This approach is relevant to the field of public health and care for vulnerable populations, providing useful evidence for community interventions. Among the most outstanding successes of the work are the use of validated instruments to measure the main variables, the statistical analysis adjusted for age and sex and the clear presentation of the results. However, the research presents deficiencies in the justification of its sample, in the critical interpretation of the findings, in the updating of the references and in the rigorous compliance with the rules of academic writing. Despite these limitations, I consider that the article has the potential to be published if the improvements outlined here are made.

introduction

The introduction adequately contextualizes the magnitude of the opioid problem at the global and local levels, highlighting the associated health, social and economic burden. The authors are right to expose the particular impact of this phenomenon in the province of cadiz, whose concentration of users justifies the investigative attention. However, the theoretical justification of the relationships between social support, quality of life and adherence remains superficial; It does not delve into the psychological mechanisms or conceptual models that sustain this relationship. In addition, the literature review includes adequate references but some are out of date and more recent studies, especially those after 2015, are not included. The hypothesis is not formally explained, although it is inferred from the objectives. I recommend clearly formulating the hypothesis and reinforcing the theoretical framework with current and specific literature on the variables involved.

method

The analytical cross-sectional design is suitable for the stated objectives. The data collection procedure, the instruments used and the sociodemographic and clinical variables considered are clearly described. The selection of instruments (whoqol-bref, duke-unc-11 and mmas-8) is a strength, since they are validated and widely used tools. However, the absence of a sample size calculation limits the interpretation of the results and weakens the inferential validity of the study. The sample, made up of 70 users recruited without probabilistic sampling, raises doubts about its representativeness and the generalisation of the findings. In addition, possible selection or information biases are not discussed, which should be included as part of the methodological limitations. Finally, although the statistical analyses are correct, the models' diagnoses are not reported, nor is the clinical relevance of the associations found discussed, beyond their statistical significance.

results

The results are presented in a clear and well structured way, with tables that facilitate their understanding. The socio-demographic and clinical description of the sample is detailed and consistent with the objectives of the study. The findings show a low quality of life, reduced social support and limited adherence to treatment, which is consistent with the literature. However, the authors limit themselves to reporting significant associations without exploring their magnitude or practical relevance in depth. It would have been desirable to include a more critical assessment of the strength of the relationships found and a discussion on the possible multicollinearity between variables.

discussion

The discussion fulfills its basic purpose, although it turns out to be superficial and uncritical. They limit themselves to repeating the results and underlining the need to strengthen social support networks, without developing a reflection on the theoretical or practical implications of the findings. In addition, the comparison with previous studies is superficial and possible discrepancies or limitations inherent to the study, such as selection bias, cross-sectional design and lack of representativeness of the sample, are not discussed. The proposed recommendations are general and not very operational, lacking concrete proposals on how to implement the suggested policies. It is recommended to broaden the discussion, compare the results with recent literature and make more specific and substantiated recommendations.

references

The references are mostly relevant, although some are out of date and more recent literature should be included to reinforce the topicality of the work.

Comments on the Quality of English Language

It must be reviewed.

Round 2

Reviewer 1 Report

Comments and Suggestions for Authors

All issues commented in the first review have been revised appropriately.